# Right Ventricle Remodelling in Left-Sided Heart Failure in Rats: The Role of Calcium Signalling

**DOI:** 10.3390/biom12111714

**Published:** 2022-11-19

**Authors:** Aleksandra Paterek, Marta Oknińska, Michał Mączewski, Urszula Mackiewicz

**Affiliations:** Department of Clinical Physiology, Centre of Postgraduate Medical Education, 01-813 Warsaw, Poland

**Keywords:** right ventricle, left-sided heart failure, myocardial infarction, calcium signalling, contractility

## Abstract

Right ventricular dysfunction (RVD) can follow primary pulmonary diseases, but the most common cause of its development is left-sided heart failure (HF). RVD is associated with HF progression, increased risk of death and hospitalisation. The mechanism of right ventricle (RV) remodelling leading to RVD due to left-sided HF is not fully elucidated. Rats underwent LAD ligation to induce extensive left ventricle (LV) myocardial infarction (MI) and subsequent left-sided HF. Sham-operated animals served as controls. After 8 weeks of follow-up, the animals underwent LV and RV catheterisation, and systolic function and intracellular Ca^2+^ signalling were assessed in cardiomyocytes isolated from both ventricles. We demonstrated that rats with LV failure induced by extensive LV myocardial infarction also develop RV failure, leading to symptomatic biventricular HF, despite only mildly increased RV afterload. The contractility of RV cardiomyocytes was significantly increased, which could be related to increased amplitude of Ca^2+^ transient, preserved SERCA2a activity and reduced Ca^2+^ efflux via NCX1 and PMCA. Our study indicates that RV failure associated with post-MI LV failure in a rat model cannot be explained by a decline in cardiomyocyte function. This indicates that other factors may play a role here, pointing to the need for further research to better understand the biology of RV failure in order to ultimately develop therapies targeting the RV.

## 1. Introduction

Right ventricular dysfunction (RVD) manifests as decreased contractility, diastolic dysfunction and right ventricular (RV) dilatation. RVD can follow primary pulmonary diseases, such as pulmonary arterial hypertension, right coronary artery infarction or genetically determined RV cardiomyopathies, such as arrhythmogenic RV dysplasia. However, the most common cause (65–80%) of RVD development is left heart disease, leading to left-sided heart failure (HF) [1,2]. 

In patients with HF, the prevalence of RVD is estimated to range from 36 to 75%, depending on the criterion adopted to qualify the right ventricle as dysfunctional (Fractional Area Change (FAC) < 45%, Tricuspid Annular Plane Systolic Excursion (TAPSE) < 15 mmHg or systolic dysfunction visualised by speckle-tracking echocardiography) [3,4,5,6]. Importantly, the presence of RVD is associated with poor prognosis and progression of HF [7], increases the risk of death and is an independent predictor of hospitalisation for HF [8,9]. RV function also plays a critical role in the selection of patients with HF for the left ventricular assist device (LVAD) implantation, as it is the RV that experiences increased stress following device implantation. It has been shown that the RV does not remodel after implantation and RV failure is the most common complication, causing significant mortality in the postoperative period [10]. It has also been shown that patients with HF who had an improvement in RV function at 40-month follow-up, as measured by TAPSE, had increased survival compared to those whose RV function deteriorated or did not change [11]. Based on these data, it seems clear that guidelines for the diagnosis of HF need to implement standardised protocols for the characterisation of RV dysfunction in order to identify the patient population at the highest risk of critical events, including death. Therefore, understanding the mechanisms of RV remodelling and RVD in left ventricular failure is key to improving the effectiveness of treatment for patients with HF. Similarly, it is important to consider the impact of therapies used in HF on RV function [12].

The mechanism of RV remodelling and development of RVD in left-sided HF is not fully elucidated. One possible mechanism that may induce RV dysfunction is an increase in RV afterload due to passive ‘backward’ transmission of pressure from LV. In both subtypes of left-sided HF, with reduced and preserved LV ejection fraction (HFrEF and HFpEF, respectively), there is an increase in left ventricular end-diastolic pressure (LVEDP). In HFrEF, impaired systolic function leads to a decrease in EF and an increase in LV late diastolic volume (LVEDV). In turn, in HFpEF, passive LV wall stiffness increases due to impaired active and passive relaxation and increased fibrosis. In both cases, LV filling is impaired and LV filling pressure increases. This may lead to a sequential increase in the left atrial (LA) pressure and an increase in pulmonary venous pressure, leading to pulmonary congestion (isolated post-capillary pulmonary hypertension (PH)). Persistent post-capillary PH may result in pulmonary arteriole remodelling, including medial hypertrophy, intimal and adventitial fibrosis and small-vessel luminal occlusion and increase in pulmonary vascular resistance (PVR) (pre-capillary PH) [13,14]. In consequence, RV afterload increases, which forces the RV to pump blood at increased pressure. This results in an increase in RV systolic wall stress, the main trigger of ventricular remodelling. Wall stress is sensed by Z-disks present in the sarcomeres of cardiomyocyte contractile apparatus [15]. This leads to the activation of pro-hypertrophic signalling, increased contractile protein synthesis, cardiomyocyte hypertrophy and RV wall thickening [16,17], which temporarily may lead to stress normalisation (adaptive hypertrophy) [18]. However, as LV failure progresses and PVR increases further, RV hypertrophy becomes insufficient to compensate for the increased wall stress. Its further increase results in RV dilatation, a decrease in ejection fraction (EF) and, eventually, inability to fill the LV (right ventricular failure (RVF)) and haemodynamic decompensation.

Another mechanism that may be responsible for RV remodelling and the development of RVD in left-sided HF is systemic neurohumoral activation. Activation of the sympathetic nervous system (SNS) and renin-angiotensin-aldosterone system (RAAS) in the short term induces a number of changes in the heart muscle, kidneys and vasculature to maintain cardiovascular homeostasis. However, with chronic activation, they exert deleterious effects, leading to fluid retention, increased myocardial preload and afterload, cardiomyocyte loss, hypertrophy, fibrosis, extracellular matrix degradation, dilatation of heart chambers and impaired intracellular Ca^2+^ signalling. Chronic neurohormonal activation in HF is now recognised as one of the most important mechanisms underlying the progression of heart failure [19].

An important determinant of both LV and RV function is cardiomyocyte contractility, which is closely dependent on changes in intracellular Ca^2+^ concentration. The functional remodelling of LV cardiomyocytes during the development of LV failure has been studied in detail in animal models and in humans, with indications that significant changes in intracellular Ca^2+^ signalling develop over time and are ultimately responsible for the decline in cardiomyocyte contractile function. It has been shown that with the development of HF, there is a decrease in Ca^2+^ transport to the sarcoplasmic reticulum (SR) by the SR Ca^2+^-ATPase (SERCA2a), an increase in Ca^2+^ efflux by the Na^2+^/Ca^2+^ exchanger (NCX1) and dysfunction of the SR Ca^2+^ channels (ryanodine receptors-RyRs), leading to diastolic Ca^2+^ leak from the SR [20,21,22]. As a consequence, SR Ca^2+^ content is reduced, contraction and relaxation kinetics are slowed and, ultimately, the amplitude of cardiomyocyte shortening is reduced. Furthermore, due to impaired Ca^2+^ signalling, the arrhythmic risk strongly increased [23]. Contrary to the detailed characterisation of cardiomyocytes isolated from the failing LV, the cardiomyocyte function and intracellular Ca^2+^ signalling has not been studied in RV exposed to the failure of LV.

Therefore, the aim of our study was to characterise RV performance in rats with LV failure following induction of LV myocardial infarction. We investigated RV function at the organ level by recording pressure-volume loops and at the cellular level. In isolated RV cardiomyocytes, we examined contractile performance, intracellular Ca^2+^ signalling and function of the main Ca^2+^ transporters. We related the results to those obtained similarly in the LV.

## 2. Materials and Methods

Male Wistar rats at 13 weeks of age (310–380 g) were used in the study in compliance with local and institutional regulations. The study conforms to the Guide for the Care and Use of Laboratory Animals, US National Institutes of Health (NIH Publication No. 85–23, revised 1996) and was approved by the local Ethics Committee (Second Warsaw Local Ethics Committee for Animal Experimentation).

### 2.1. Experimental Design

Rats were subjected to induction of the LV myocardial infarction (MI) (HF group) or sham operation (Sh group). After 8 weeks of follow-up, the animals underwent echocardiographic imaging to evaluate the size of MI. Only rats with large MI (≥40% of the LV) were enrolled in the HF group in this study. Subsequently rats underwent LV and RV and catheterization. Ketamine HCl/xylazine overdose was given to euthanize the animals and hearts were excised for LV and RV cardiomyocyte isolation.

### 2.2. Induction of the Left Ventricle Myocardial Infarction

The animals were anaesthetized by intraperitoneal administration of a mixture of ketamine (100 mg/kg body weight) and xylazine (5 mg/kg body weight). Subsequently, the chest was opened and a 6-0 suture was put around the proximal left anterior descending artery. It was tightened in the MI group, but in the Sh group it was removed. Subsequently, the chest was closed and the pneumothorax was evacuated. Approximately 1 h after the surgery a single dose of an antibiotic narcotic analgesic agent was given intraperitoneally. The procedure was described previously in detail [20].

### 2.3. Echocardiographic Imaging

Echocardiography was performed using MyLab25 (Esaote, Italy) with 13 MHz linear array transducer. Under light anaesthesia (ketamine HCl and xylazine, 75 mg and 3.5 mg/kg body weight, IP) LV end-diastolic and end-systolic dimensions were determined from the short-axis view at the midpapillary level and long-axis view. Regional LV wall motion abnormalities were quantitated as described previously. Contractility of 12 wall segments visualized in the midpapillary short-axis view and 11 segments visualized in the long-axis view were graded as 1 (normal) or 0 (abnormal) and the total wall motion score index (WMI) was calculated. The normal hearts have WMI = 23. Our previous results revealed that WMI closely correlated with infarct size and that WMI = 15 corresponded to infarct size ~40% [24].

### 2.4. Hemodynamic Measurements in LV and RV—Pressure-Volume Loops

Rats were put on a heating pad, anaesthetized with ketamine and xylazine, intubated and put on an animal ventilator. The upper abdominal cavity was opened and through the cutting of the diaphragm, the heart was exposed. The left and subsequently right ventricular apex was punctured with a 25 G needle and a microtip P–V catheter (SPR-838, Millar Instruments; Houston, TX, USA) was inserted into the LV and RV. Its position was established based on pressure and volume signals. After stabilization for 5 min, the signals were continuously recorded at a sampling rate of 1000/s using an ARIA P–V conductance system (Millar Instruments) coupled to a PowerLab/4SP A/D converter (AD Instruments; Mountain View, CA, USA) and a personal computer. To characterize cardiac function, first the inferior vena cava was compressed for 10 s and then released to achieve reduction and augmentation of venous return and cardiac preload, respectively. Heart rate, maximal systolic pressure (ESP), end-diastolic pressure (EDP), maximal slope of systolic pressure increment (+dP/dt max) and maximal slope of diastolic pressure decrement (−dP/dt max), ejection fraction (EF), end-diastolic volume (EDV), end-systolic volume (ESV), stroke volume (SV), and cardiac output (CO) were computed using a cardiac P–V analysis program (PVAN3.2, Millar Instruments) for both ventricles. Indexes of contractility and stiffness (slope of end-systolic and end-diastolic P–V relations (ESPVR and EDPVR)) were also calculated using PVAN3.2.

### 2.5. Cardiomyocyte Isolation

The heart was rapidly excised and perfused for 10 min with nominally Ca^2+^ free Tyrode’s solution containing 100 mmol/l EGTA. The initial washout period was followed by 20 min of perfusion with Ca^2+^ free Tyrode’s solution containing 20 mg collagenase B (Roche) and 3 mg protease (Sigma) per 30 mL. Thereafter, the right and left ventricles were separated, placed in the glass dishes and mechanically disrupted. The cell suspension was filtered and allowed to sediment. The supernatant was discarded and cells were washed twice with Tyrode’s solution. The Ca^2+^ concentration was increased gradually to 1.8 mmol/L.

### 2.6. Recording of Ca^2+^ Transient and Evaluation of the Activity of Cardiomyocyte Ca^2+^ Transporters

The myocytes were incubated for 20 min with 10 μM Indo-1 and perfused at 37 °C with a Tyrode solution containing 1.8 mmol/L Ca^2+^. The difference between the systolic and diastolic Indo-1 fluorescence (excited at 365 and measured as ratio of fluorescence at 405 and 495 nm) was used as a measure of the amplitude of Ca^2+^ transients (Dual-Channel Ratio Fluorometer, Biomedical Instrumentation Group, University of Pennsylvania). The rate of Ca^2+^ transport by sarcoplasmic reticulum (SR) Ca^2+^-ATPase (SERCA2a), Na^+^/Ca^2+^ exchanger (NCX1) and plasma membrane Ca^2+^-ATPase (PMCA) was estimated from the rate constants (r1, r2, r3) of the single exponential curves fitted to electrically and caffeine-evoked Ca^2+^ transient decay [25].. The rate constants of the Ca^2+^ transient decay for SERCA2a and NCX1 were calculated according to formulas: r_SERCA_ = r1−r2 and r_NCX_ = r2−r3, respectively, while r3 was taken as the measure of the rate of Ca^2+^ transport by PMCA (r3 = r_PMCA_). rSERCA, rNCX and rPMCA describe the average velocity of Ca^2+^ transport. SR Ca^2+^ content was estimated from the amplitude of caffeine-evoked Ca^2+^ transients in myocytes superfused with Na^+^-, Ca^2+^-free (0Na0Ca) solution. ISO2 Multitask-Patch-Clamp Software (University of Dresden) was used to fit mono-exponential curves to decaying part of Ca^2+^ transients and calculate *r*1, *r*2 and *r*3 rate constants.

### 2.7. Recording of Cardiomyocyte Shortening

Myocyte contractions were recorded by a video-edge detection system including a fast digital dimensioning video camera (IonOptix LLC, Milton, MA, USA) enabling acquisition of the cell length changes in real time. Myocyte contractions were elicited by electrical pacing at 1 Hz. Cell shortening was taken as a measure of myocyte contractile performance. The amplitude and time course of the signal were analysed by the IonWizard software (IonOptix). Contraction amplitude was expressed as the difference between systolic and diastolic cell length and normalized as a percentage of the resting cell length. The contraction time (time to peak) was calculated as the time from the initiation of contraction to the maximal cell shortening. The time required for re-lengthening cell to 90% of the resting cell length was taken as the relaxation time. The maximal rate of contraction and relaxation was evaluated to describe the kinetics of contraction and relaxation process.

### 2.8. Statistical Analysis

Normal data distribution was verified by the Shapiro–Wilk test. For normally distributed data, two-way ANOVA followed by the Student–Newman–Keuls as post hoc test. In case of a lack of normality, the Kruskal–Wallis test and Dunn’s post hoc test were used for all pairwise comparisons. Differences between the 2 groups were detected using the Mann–Whitney Rank Sum test or t-student test, depending on the normality of data distribution (Sigma Plot v.11.0). Differences were considered significant at a level of *p* < 0.05.

## 3. Results

### 3.1. Left and Right Ventricular Function

An average MI size was 42.5% in the HF group (Table 1). Pressure-volume loop analysis revealed that 8 weeks after MI induction, rats in the HF group developed: (i) LV systolic dysfunction as evidenced by reduced LV EF (Figure 1A), impaired contractility (indicated by both reduced +dP/dt max (Figure 1F) and ESPVR (Figure 2B) and, consequently, reduced CO (Figure 1B), (ii) LV dilation, indicated by increased LV EDV (Figure 1C) and (iii) LV diastolic dysfunction reflected by a reduced −dP/dt max (Figure 1G).

Since arterial elastance (Ea), a marker of arterial component of LV afterload, was increased and contractility was impaired, a very sensitive index of LV ventricular–arterial coupling (Ea/ESPVR) was increased by more than five-fold, indicating severe uncoupling (Figure 2C). Markedly increased lung weight, a marker of lung congestion, indicated symptomatic LV heart failure (Table 1), accompanied by reduced LV ESP (Figure 1D) and CO (Figure 1B) and elevation of LV EDP (Figure 1E). Moreover, LV hypertrophy was found, as indicated by LV weight (Table 1).

RV systolic function was also reduced, which was evidenced by impaired RV EF (Figure 1A), +dP/dt max (Figure 1F) and ESPVR (Figure 2B), though the magnitude of RV dysfunction was lower as compared to LV. RV dilation was indicated by increased EDV (Figure 1C) while RV hypertrophy by increased RV weight (Table 1). Pulmonary Ea increased by approximately 30%, reflecting increased pulmonary vascular resistance, resulting in increased RV ESP (Figure 1D). Due to increased Ea and reduced ESPVR, Ea/ESPVR was doubled, indicating ventricular–arterial uncoupling, but again, the magnitude of that uncoupling was significantly lower than for LV (Figure 2C). All HF rats presented with either pleural effusion or ascites or both, indicating RV failure. In summary, the HF rats demonstrated biventricular HF, with more pronounced LV dysfunction and mildly increased PVR. Representative RV pressure and volume signals and pressure-volume loops for RV and LV are shown in Figure 2D,E.

### 3.2. Calcium Signalling

Calcium signalling was studied in cardiomyocytes isolated separately from LV and RV in HF and sham rats, according to the protocol shown in Figure 3A (see also Methods section). In cardiomyocytes isolated from the LV 8 weeks after MI induction, amplitude of Ca^2+^ transient was slightly increased compared to that in cardiomyocytes from sham-operated animals (Figure 3B,C), while the SR Ca^2+^ content was unchanged; however, a trend towards lower Ca^2+^ content in the SR was evident (Figure 3E). This suggests an increase in SR Ca^2+^ fractional release, which is probably a consequence of the greater sensitivity of RyRs to Ca^2+^, due to their increased phosphorylation and oxidation, a phenomenon often observed in heart failure.

The rate of Ca^2+^ transient decay in the LV cardiomyocytes was reduced (Figure 3D) due to decreased activity in SERCA2a responsible for SR Ca^2+^ reuptake (Figure 3F) and PMCA activity removing Ca^2+^a from the cell (Figure 3H). The decrease in SERCA2a activity results in a decrease in Ca^2+^ transport to the SR. However, a decrease in Ca^2+^ efflux through PMCA leads to intracellular Ca^2+^ retention, exposing SERCA2a to higher Ca^2+^ concentrations, which, to some extent, can compensate for the decrease in its activity and sustain Ca^2+^ transport to the SR. Hence, the SR Ca^2+^ content is only slightly reduced. NCX1 function at this stage of HF development was unchanged in LV cardiomyocytes (Figure 3G).

In the RV cardiomyocytes isolated from rat hearts after induction of left ventricular MI, amplitude of Ca^2+^ transient was significantly higher than in the RV cardiomyocytes of sham-operated animals. The increase in amplitude was twice as high in RV cardiomyocytes as in the LV cardiomyocyte surviving after MI (30% in RV vs. 15% in LV) (Figure 3B,C). The SR Ca^2+^ in RV cardiomyocytes was unchanged, but there was a clear trend towards its increase (*p* = 0.08) in post-MI animals compared to sham-operated ones (Figure 3E). SERCA2a activity in the RV cardiomyocytes, in contrast to the LV cardiomyocytes, was preserved and even showed an increasing trend (Figure 3F), while the activity of both membrane transporters, NCX1 and PMCA, was significantly reduced (Figure 3G and H, respectively). This led to intracellular Ca^2+^ retention, exposing SERCA2a to higher Ca^2+^ concentrations, which, combined with its preserved activity, results in sustained SR Ca^2+^ content and significantly increases Ca^2+^ transient amplitude. The rate of Ca^2+^ transient decay was unchanged in the RV cardiomyocytes from failing hearts (Figure 3D), which was consistent with preserved SERCA2a activity.

### 3.3. Myocyte Shortening

Representative recordings of cell shortening in cardiomyocytes isolated from Sh and MI rats are presented in Figure 4A. The amplitude and the contraction and relaxation rates in cardiomyocytes isolated from the LV after MI induction were unchanged compared to LV myocytes of sham-operated rats (Figure 4B). We observed only a slight prolongation of time to peak contraction, which may suggest a disturbance in electromechanical coupling, resulting in impaired Ca^2+^ release from the SR (Figure 4C).

In contrast, in RV cardiomyocytes isolated from rats after induction of LV infarction, we found both an enhancement of the amplitude of myocyte shortening and an increase in kinetics of the contraction–relaxation cycle. The amplitude of the myocyte shortening was increased by approximately 25%, which corresponds well with the increase in Ca^2+^ transient amplitude in these cardiomyocytes (Figure 4B). In addition, the rate of contraction and relaxation increased by 30% and 43%, respectively (Figure 4E,F, respectively). Therefore, the duration of relaxation was unchanged, despite the significantly increased amplitude of myocyte shortening (Figure 4D). However, the increase in contraction kinetics was not sufficient to compensate for the increase in time to contraction peak (Figure 4C). However, the total contraction–relaxation cycle duration was not prolonged, as compared to that measured in sham-operated animals (229 ± 6 ms in MI rats vs. 217 ± 9 in sham rats, *p* = 0.276).

## 4. Discussion

Here, we show that rats with LV failure induced by a large LV myocardial infarction develop also RV failure, resulting in symptomatic biventricular HF, despite only mildly increased RV afterload. Furthermore, to our best knowledge, we are the first to show that contractility of RV cardiomyocytes is increased in this model due to preserved SERCA2a function and intracellular Ca^2+^ retention, resulting from reduced Ca^2+^ efflux through sarcolemmal Ca^2+^ transporters, NCX1 and PMCA. This suggests that RV failure in the rat model of post-MI LV failure can neither by explained by extensive pulmonary hypertension and the resulting increase in RV afterload nor by RV cardiomyocyte dysfunction.

### RV Failure as Complication of LV Failure

Here, we show that 8 weeks after induction of MI, rats develop typical systolic LV failure with markedly reduced contractility, LV dilation, elevated LVEDP, ventricular–arterial uncoupling due to markedly reduced LV contractility/mildly increased arterial elastance (Ea, a marker of arterial part of ventricular afterload) and pulmonary congestion. It is accompanied by RV failure, as evidenced by reduced RV contractility, RV dilation and ventricular–arterial uncoupling due to both increased Ea (caused by increased PVR) and reduced contractility; however, the increase in RV Ea was quite mild (approx. 30%) as compared to that in the models of monocrotaline-induced pulmonary hypertension and pulmonary artery banding, where three–five-fold increases were found [26,27]. Model RV is able to generate much higher systolic pressures and typically initially goes through the phase of at least a temporary compensatory increase in contractility, decompensating when Ea is increased by many fold, at much higher afterloads. Therefore, there must be something specific about RV failure related to left-sided HF, which facilitates RV decompensation. To answer this question, we looked at function of cardiac myocytes isolated from both ventricles.

In LV cardiomyocytes, the amplitude of myocyte shortening was preserved. However, the duration of the systolic–diastolic cycle was prolonged. In our previous papers, we obtained similar results in cardiomyocytes after induction of a large MI of LV [20,28]. Maintenance of contractility of the remaining cardiomyocytes is probably necessary to compensate for the massive loss in cardiomyocytes after large LV infarction as well as for significant and adverse LV dilation and maintains ventricular systolic function. However, prolongation of the contraction–relaxation cycle impairs LV filling during diastole and increases duration of systole, impairing the conditions for effective coronary perfusion and facilitating hypoxia [29].

Moreover, in LV cardiomyocytes, we observed the well-known major signs of cellular remodelling of failing heart [21,22]. SERCA2a transporting activity was significantly reduced, which translated into a trend towards lower Ca^2+^ content in the SR. In spite of that, the amplitude of the Ca^2+^ transient was increased, which indicated enhanced fractional release. It suggests a higher sensitivity of RyRs to Ca^2+^ ions, which is strictly dependent on their phosphorylation and oxidation level [30]. However, the compensatory increase in SR Ca^2+^ release comes at the expense of diastolic Ca^2+^ leak from the SR, which acutely leads to a decrease in the SR Ca^2+^ content and, consequently, to a decrease in cardiomyocyte contractility and haemodynamic decompensation and increases the propensity to arrhythmias. In our model, at this stage of HF development, the Ca^2+^ transient amplitude is still amplified. This maintains the contractility of the LV cardiomyocytes and prevents complete LV decompensation.

In contrast, in cardiomyocytes isolated from the RV, despite its marked haemodynamic dysfunction, no such changes were apparent. SERCA2a function was preserved. The SR Ca^2+^ content showed an increasing trend, which corresponded to increased Ca^2+^ transient amplitude and suggested normal RyRs function without increased diastolic Ca^2+^ leak from the SR. The main change observed at the level of Ca^2+^ signalling in RV cardiomyocytes was a marked decrease in sarcolemmal Ca^2+^ efflux by NCX and PMCA. This interesting compensatory mechanism leads to intracellular Ca^2+^ retention, which exposes the SERCA2a to higher Ca^2+^ concentrations and sustains its function and ultimately translates into higher amplitude of Ca^2+^ transient. Further, the amplitude of myocyte shortening was significantly increased and the duration of contraction and relaxation was not prolonged due to the preserved function of SERCA2a and the increased maximum rate of the contraction and relaxation. These results suggest that in this model, the RV is at a different stage of cellular remodelling than the LV.

Interestingly, in our previous study, also in the LV cardiomyocytes at an early stage of remodelling (3 days after MI induction), we showed an increase in Ca^2+^ transient amplitude due to a decrease in NCX1 function. Furthermore, we showed that the decrease in NCX1 transport activity in LV cardiomyocytes is associated with calcineurin activation and is abolished by its blockade by cyclosporine [31]. In addition, Katanosaka et al [32] showed that calcineurin inhibits NCX1 in neonatal phenylephrine-treated rat cardiomyocytes by directly binding its C-terminus to the beta-1 repeat of the central cytoplasmic loop of NCX1. Calcineurin is anchored to the Z-discs of the contractile apparatus via the MLP protein [33]. Thus, it appears that increased RV afterload may be sensed by the Z-disks of the contractile apparatus, which are mechanosensors of stress in the myocardial wall [15] and lead to activation of calcineurin and subsequent inhibition of NCX1 function.

Another mechanism responsible for the decrease in NCX1 activity in RV myocytes may be the regulation of the expression of proteins involved in intracellular Ca^2+^ signalling by miRNAs [34]. Van Rooij et al. [35] showed that miRNA-214 is sensitive to cardiac stress and is upregulated in cardiac hypertrophy and failure. Increased RV afterload in our model and RV hypertrophy may lead to activation of miRNA-214 and inhibition of NCX1 expression. Indeed, Aurora et al. [36] showed that the NCX1 mRNA (encoded by Slc8a1) had three conserved miRNA-214-binding sites in the 3′-UTR, and the NCX1 protein expression significantly increased in the myocardium of miRNA-214 KO-mice versus wild-type mice. In summary, in this model, modestly increased RV afterload may be sufficient to activate calcineurin and miR-214 and result in inhibition of NCX1 responsible for the compensatory increase in RV myocyte contractility.

However, despite enhanced RV cardiomyocyte contractility, RV dysfunction is present. We can only speculate that there is a reduction in the number of cardiomyocytes in the RV by their gradual loss (vs. sudden loss in the LV) through apoptosis and/or necrosis, followed by replacement with fibrous tissue. Perhaps this is due to inadequate oxygen supply, resulting from pulmonary congestion and impaired gas exchange. In our previous work, we showed that a decrease in the O_2_ partial pressure also occurs in the RV in monocrotaline-induced pulmonary hypertension [27]. RV afterload in this left-sided HF model is not as high as in the pulmonary hypertension model, but in combination with low oxygen supply and enhanced contractility of RV cardiomyocytes, it may lead to hypoxia. An additional indication that hypoxia may lead to cardiomyocyte loss is provided by the results of our earlier work, investigating the effect of myo-inositol trispyrophosphate (ITPP), an allosteric effector of haemoglobin, on post-myocardial LV remodelling. ITPP increases the oxygen-releasing capacity of red blood cells and increases oxygen supply to the tissues. In a model of post-MI HF, it improved LV function without affecting the contractility of individual cardiomyocytes [37]. Hence, it can be concluded that alleviation of myocardial tissue hypoxia by ITPP may inhibit the loss of working cardiomyocytes.

Another factor responsible for the development of RV dysfunction despite the enhanced contractility of its individual cardiomyocytes may be impaired synchronisation of myocyte contraction due to disturbances in propagation of the depolarization wave. Impaired electrical impulse conduction may occur due to decreased expression and localisation of intercellular gap-junction channels or enhanced myocardial fibrosis. Increased connective tissue deposition between cardiomyocytes separates them, impeding the formation of connexons and complicating the conduction pathway, which prolongs the depolarisation time of the ventricle and reduces synchronization of contractile activity of individual cardiomyocytes [38]. Moreover, disruption of the cytoskeletal proteins responsible for the mechanical cardiomyocyte coupling in desmosomes can also lead to abnormal coordination of individual cardiomyocyte contraction and a decrease in RV haemodynamic performance [39].

Based on observations of the LV undergoing remodelling after extensive MI, ultimately leading to the development of HF, changes in intracellular Ca^2+^ signalling increasing cardiomyocyte contractility are transient. In the LV cardiomyocytes, the increase in Ca^2+^ transient amplitude due to the decline in NCX1 function was short lived and disappeared over time after MI induction and was replaced by a progressive decline in SERCA2a activity [31]. If an analogous mechanism occurs in the RV, it is likely to lead to further progression of RV failure and complete haemodynamic decompensation of the heart. Understanding the mechanisms that maintain RV systolic function by protecting the contractility of its cardiomyocytes and preventing their potential loss may improve the prognosis and survival of patients with left-sided HF.

## 5. Conclusions

Our study indicates that RV failure associated with post-MI left-sided HF in a rat model cannot be explained by abnormalities in cardiomyocytes. Furthermore, a compensatory increase in cardiomyocyte contractility cannot prevent the development of HF. This points to the need for further research to better understand the biology of RV failure, with the consequent development of therapies for this disease.

## Figures and Tables

**Figure 1 biomolecules-12-01714-f001:**
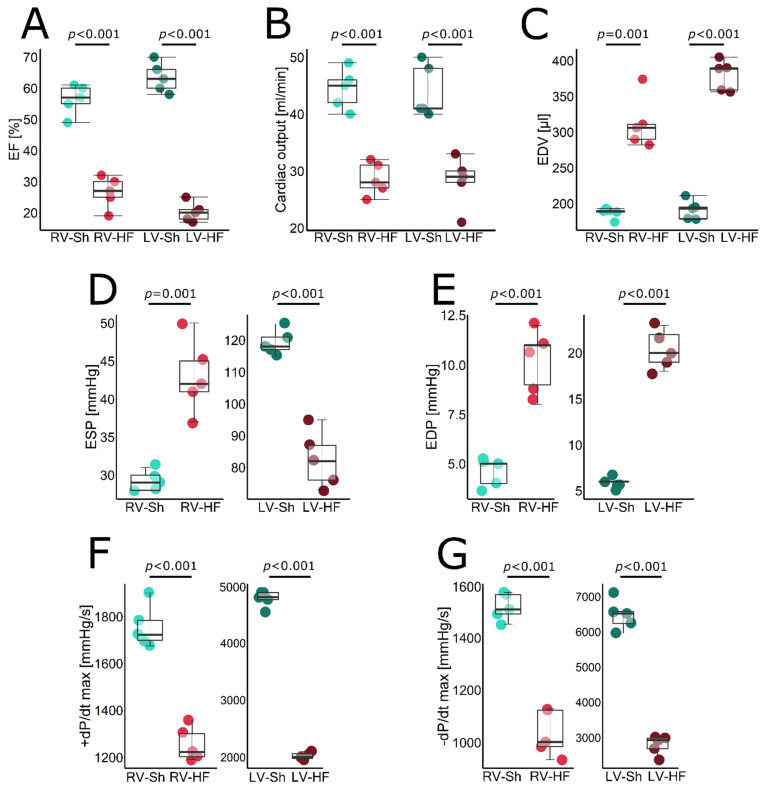
Pressure-volume-loop-derived hemodynamic parameters in LV and RV of sham-operated (Sh) and heart-failure (HF) rats. (**A**) Ejection fraction (EF), (**B**) cardiac output (CO), (**C**) end-diastolic volume (EDV), (**D**) end-systolic pressure (ESP), (**E**) end-diastolic pressure (EDP), (**F**) maximal slope of systolic pressure increment (+dP/dt max), (**G**) maximal slope of diastolic pressure decrement (−dP/dt max). Data presented as median, Q1, Q3, max and min values.

**Figure 2 biomolecules-12-01714-f002:**
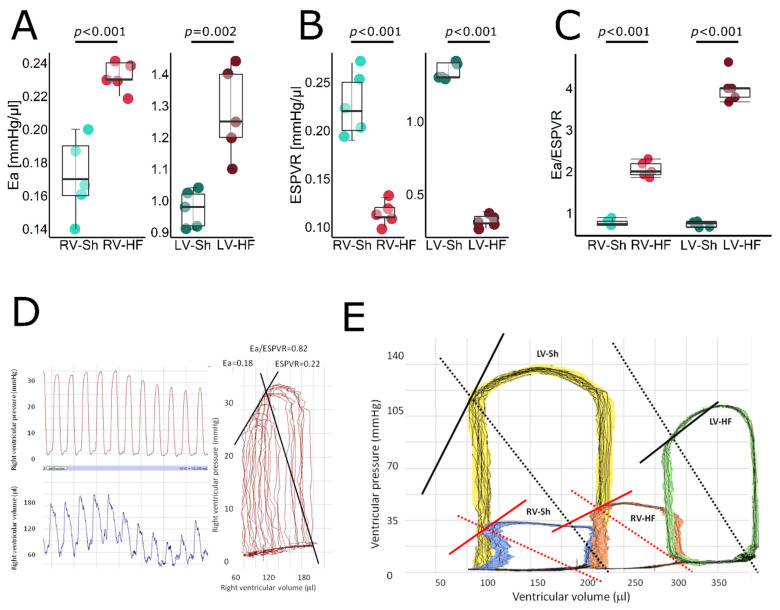
Pressure-volume loop analysis—stiffness, contractility and ventricular–arterial coupling in LV and RV of control and failing hearts. (**A**) Arterial elastance (Ea), (**B**) end-systolic pressure-volume relation (ESPVR), (**C**) ventricular-arterial coupling index (Ea/ESPVR), (**D**) representative LV pressure signals (left upper panel), LV volume signals (left lower panel) and reconstructed pressure-volume loops from these signals (right panel) from a sham-operated rat. (**E**) Representative pressure-volume loops obtained from (left to right) left ventricle of sham-operated rats (the yellow loop), right ventricle of sham-operated rats (blue loops), right ventricle of failing heart (orange loop) and left ventricle of failing heart (green loop). Solid line represents ESPVR, dashed line represents Ea. Data presented as median, Q1, Q3, max and min values.

**Figure 3 biomolecules-12-01714-f003:**
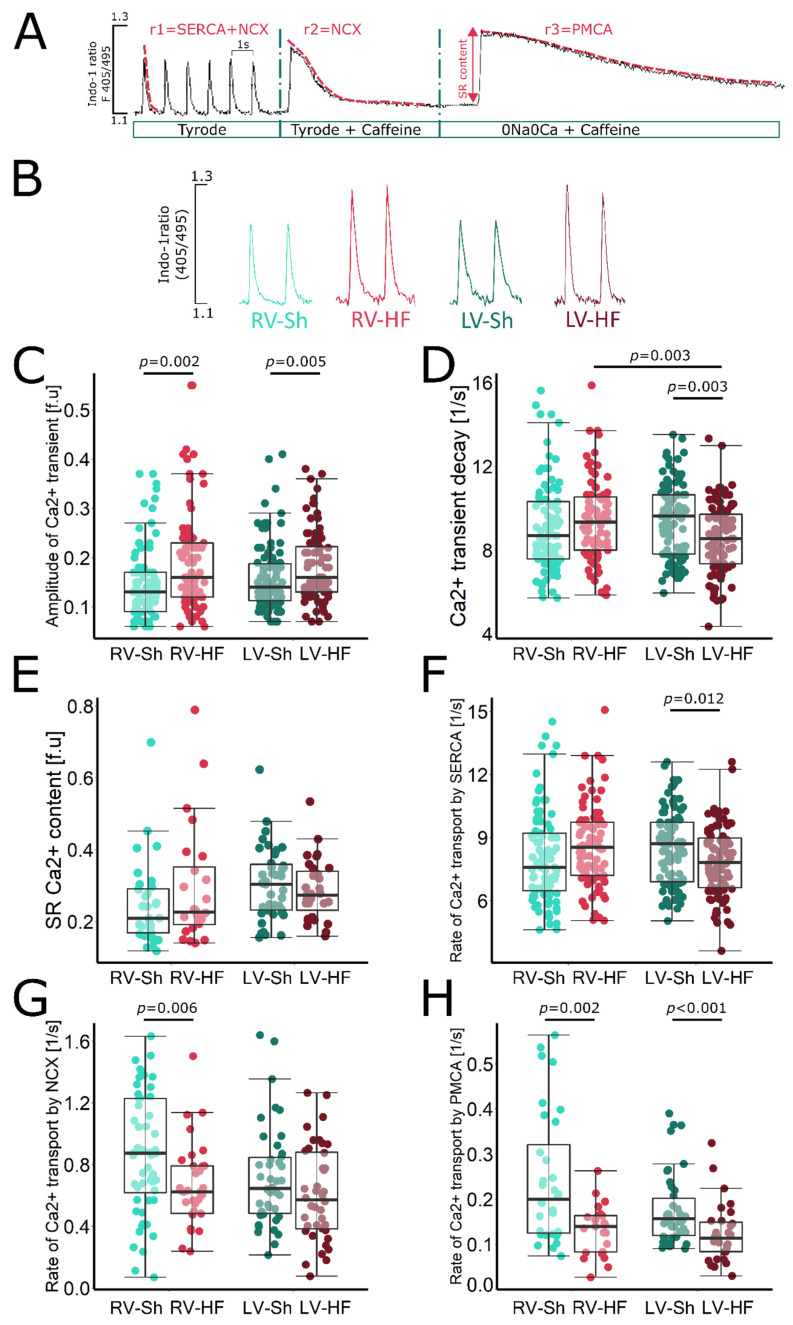
Calcium signalling in cardiomyocytes isolated from LV and RV of sham-operated (Sh) and heart-failure (HF) rats. (**A**) Experimental protocol used to assess Ca^2+^ signalling in cardiomyocytes: cardiomyocytes were superfused with Tyrode solution and were stimulated at 1 Hz, caffeine was applied to cardiomyocytes superfused with Tyrode or Na^+^/Ca^2+^ free solution (0Na0Ca). Exponential curves were fitted to decaying part of electrically or caffeine-evoked Ca^2+^ transients and rate constants of their decay (r1, r2 and r3) were calculated. The rate of Ca^2+^ transport by SERCA2a, NCX1 and PMCA was calculated according to formulas: rSERCA = r1–r2, rNCX = r2–r3 and rPMCA = r3. (**B**) Representative original recording of electrically evoked Ca^2+^ transients from cardiomyocytes isolated from left and right ventricles of healthy and failing hearts. (**C**) Amplitude of Ca^2+^ transient, (**D**) rate of Ca^2+^ transient decay (r1) in electrically stimulated myocytes, (**E**) sarcoplasmic reticulum (SR) Ca^2+^ content, (**F**) rate of Ca^2+^ transport by SERCA2a (rSERCA = r1–r2), (**G**) rate of sarcolemmal Ca^2+^ efflux (r2), (**H**) rate of Ca^2+^ transport by NCX1 (rNCX = r2–r3) and PMCA (rPMCA = r3), respectively. Data presented as median, Q1, Q3, max and min values, *n* = 10–25 cardiomyocytes per heart, 5 hearts per group.

**Figure 4 biomolecules-12-01714-f004:**
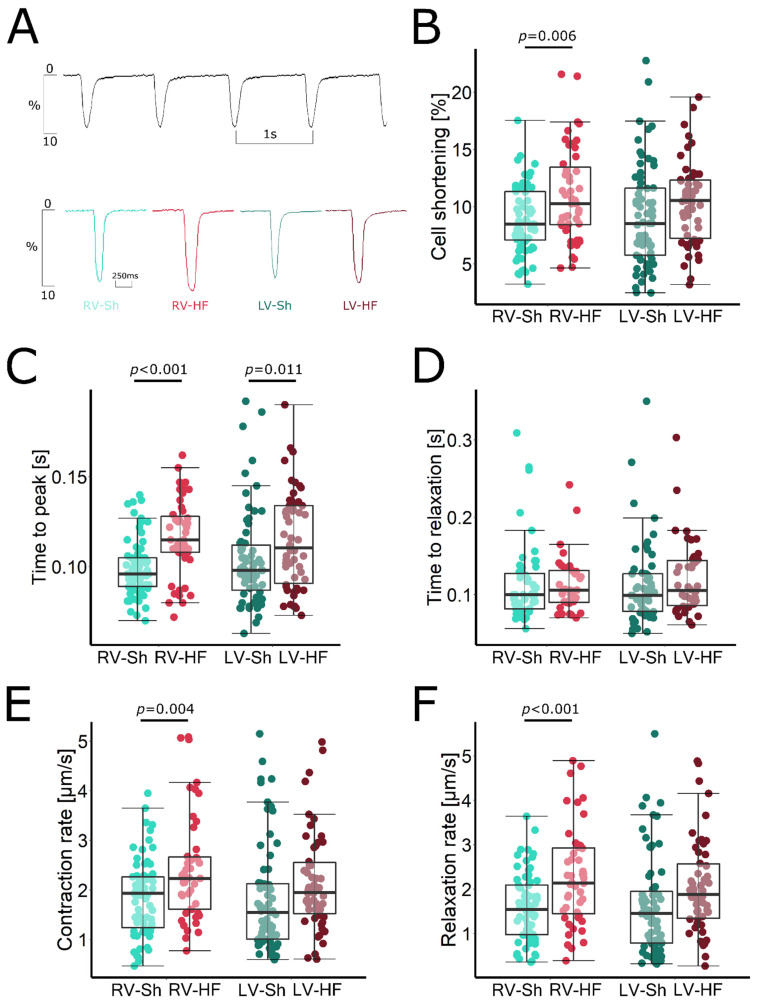
Contractile function of cardiomyocytes isolated from LV and RV of sham-operated (Sh) and heart-failure (HF) rats. (**A**) Representative recordings of sarcomere shortening, (**B**) amplitude of sarcomere shortening, (**C**) time to peak contraction, (**D**) time to 90% of relaxation, (**E**) contraction rate, (**F**) relaxation rate. Data presented as median, Q1, Q3, max and min values, *n* = 10–20 cardiomyocytes per heart, 5 hearts per group.

**Table 1 biomolecules-12-01714-t001:** Morphological parameters.

Parameter	Sham (*n* = 5)	SEM	HF (*n* = 5)	SEM	*p*-Value
BW, g	443	17	432	17	0.66
HW/TL, g/cm	0.26	0.01	0.48	0.07	0.006
LV/TL, g/cm	0.175	0.004	0.234	0.002	0.04
LV, g	0.68	0.02	0.82	0.05	0.02
RV/TL, g/cm	0.043	0.0023	0.112	0.0192	0.01
RV, g	0.17	0.01	0.35	0.05	0.001
RV/LV	0.25	0.013	0.48	0.081	0.02
Lungs/TL, g/cm	0.44	0.02	1.03	0.14	0.01
Dry Lungs/TL, g/cm	0.01	0.003	0.22	0.03	0.01
Liver/TL, g/cm	3.09	0.2	3.96	0.3	0.03
MI size (%)	-	-	42.5	1.6	-

BW—Body weight, HW—heart weight, TL—Tibia length, RV—right ventricle, LV—left ventricle.

## Data Availability

The datasets are available from the corresponding author upon request.

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
