# Peer review of "Right Ventricle Remodelling in Left-Sided Heart Failure in Rats: The Role of Calcium Signalling"

_biomolecules, 2022, doi:10.3390/biom12111714_

Round 1

Reviewer 1 Report

Authors present quite interesting article on the interplay between LV and RV heart failure. However, I have two major concern related to Ca transient measurements.

1. Figure 3 shows Ca transients in f.u., but some how SR content was estimated in nM, which would require a calibration. Please, describe calibration in the methods.

2. SR content value is too low. Typical value should be at least 100 µM. If calibration to achieve concentration units is problematic, I would suggest to use f.u. (same as Ca transient amplitude).  

Author Response

Response to Reviewer 1 Comments

Authors present quite interesting article on the interplay between LV and RV heart failure. However, I have two major concern related to Ca transient measurements.

  1. Figure 3 shows Ca transients in f.u., but some how SR content was estimated in nM, which would require a calibration. Please, describe calibration in the methods.
  2. SR content value is too low. Typical value should be at least 100 µM. If calibration to achieve concentration units is problematic, I would suggest to use f.u. (same as Ca transient amplitude).  

Thank you very much for pointing out this mistake.  As suggested, we expressed the SR Ca2+ content in fluorescence units (Fig. 3E), analogous to the amplitude of the Ca2+ transient (Fig. 3C).

Reviewer 2 Report

This is a very well designed and conducted study investigated the potential causation of RV failure induced by LV MI, and the results of preserved Ca2+ regulating function along with increased cardiomyocyte contractility in failed RV overturned the previous understanding of the cause to RV failure, the simply deteriorated and abnormal RV cardiomyocytes. This study opened a new track for a deep understanding of the mechanism of RV failure, and shed lights for new interventions targeting on HF. The manuscript is largely well-written, here are some of my concerns.

1.       For the animal model, the Authors only mentioned the body weight of the rats used, not the age, which is a less precise way of picking the experimental subjects, even for animals. Age is a huge factor that can enormously affect heart function and morporlogy of myocardium, although a similar body weights might be seen. Please specify the age of the rats used in the study.

2.       In the morphological data, only normalized ventricle data are shown in the table. Original mass of RV and LV would also be valuable to show here to give direct sense to the readers.

3.       Clearly there is a hypertrophy of the failed RV and LV, any pictures to visually show it? Or any histo-staning to show this morphological change?

4.       All data presented for the heart function in the study are physiological performance data, any biochemical evidence for the sham and HF groups to show the heart failure, i.e. BNP or NT-proBNP?

5.       If the function of Ca2+ regulating proteins is changed in RV failure, any data for the protein content level? The activity alteration of proteins, especially enzymes, needs to firstly rule out the effect of protein content change.

6.       The force output is also a critical factor to evaluate myocardium function, but in the contractility data, there is no such measurement, it would be good to have the force data with the other contractile parameters.

7.       Check all of the abbreviations, make sure there is a full name when it firstly appears in the text.

8.       What equipment was used to read the Ca2+ transient signal? Also the method in this part needs more details or maybe some references are also needed, as this is helpful for readers to understand the data figure, and this is almost the most important data of the whole study.

9.       The second sentence of section 2.6 was not grammarly correct

10.   In page 8 first paragraph, it should be “RyR”

Author Response

Response to Reviewer 2 Comments

This is a very well designed and conducted study investigated the potential causation of RV failure induced by LV MI, and the results of preserved Ca2+ regulating function along with increased cardiomyocyte contractility in failed RV overturned the previous understanding of the cause to RV failure, the simply deteriorated and abnormal RV cardiomyocytes. This study opened a new track for a deep understanding of the mechanism of RV failure, and shed lights for new interventions targeting on HF. The manuscript is largely well-written, here are some of my concerns.

  1. For the animal model, the Authors only mentioned the body weight of the rats used, not the age, which is a less precise way of picking the experimental subjects, even for animals. Age is a huge factor that can enormously affect heart function and morporlogy of myocardium, although a similar body weights might be seen. Please specify the age of the rats used in the study.

Thank you very much for this good point. In the study we used male rats of 13 weeks of age. This information was included in the revised manuscript (Chapter 2, first sentence).

  1. In the morphological data, only normalized ventricle data are shown in the table. Original mass of RV and LV would also be valuable to show here to give direct sense to the readers.

We have added the original mass of RV and LV in Table 1 as suggested by the Reviewer.

  1. Clearly there is a hypertrophy of the failed RV and LV, any pictures to visually show it? Or any histo-staning to show this morphological change?

Thank you very much for this comment. Unfortunately, we do not have such data because whole hearts after haemodynamic measurements were taken from the rats and enzymatically digested to obtain cardiomyocytes for functional measurements.

  1. All data presented for the heart function in the study are physiological performance data, any biochemical evidence for the sham and HF groups to show the heart failure, i.e. BNP or NT-proBNP?

Thank you very much for this remark. Unfortunately, we did not determine biochemical markers of HF in this study. The presence of symptomatic HF in this model was confirmed by a dramatic decrease in EF (about 20%), an increase in LVEDP to about 20 mmHg (whereas in the rat an increase in LVEDP above 14 mmHg confirms the presence of HF) and an almost threefold increase in the ratio of lung weight to tibia length. However, in our earlier work (Kalisz et al, 2015; PMID: 26579573) in the same model of HF in rats, 8 weeks after induction of large left ventricular MI, we confirmed the presence of HF also by biochemical methods showing an approximately 3-fold increase in BNP expression at the mRNA level in LV tissue with respect to sham operated animals.

  1. If the function of Ca2+ regulating proteins is changed in RV failure, any data for the protein content level? The activity alteration of proteins, especially enzymes, needs to firstly rule out the effect of protein content change.

In this work, we measured the rate of Ca2+ transport by the major calcium transporters in cardiomyocytes. We fully agree that the changes we have shown in the rate of Ca2+ transport by these transporters may depend on both changes in their expression at protein level and changes in their intrinsic activity. However, it is the change in the rate of Ca2+ ion transport that determines the cardiomyocyte contractile function, regardless of the cause of this change (change in expression and/or function of the transporting proteins). In our earlier work (Mackiewicz et al. 2009; DOI: doi.org/10.1093/cvr/cvn285) in the same rat model of post-MI HF, we showed that SERCA2a expression at the protein level is not altered in LV of HF rats compared to sham operated rats, suggesting that the decrease in SERCA2a transport activity is due to a decrease in its activity rather than expression in this model. However, we agree that investigating changes in the content of proteins involved in the cardiomyocyte Ca2+ transport in the RV and LV would be a good complementation of our results.

  1. The force output is also a critical factor to evaluate myocardium function, but in the contractility data, there is no such measurement, it would be good to have the force data with the other contractile parameters.

We absolutely agree with this comment. Measurement of the maximal force generated in isovolumic contraction of the cardiomyocyte would be an excellent addition to the description of RV and LV cardiomyocyte contractile function. Unfortunately, we do not have the equipment to make such a measurement and were therefore limited to a method of measuring cardiomyocyte contractile function by edge detection system (IonOptix, USA) enabling acquisition of the cell length changes in real-time (cell shortening).

  1. Check all of the abbreviations, make sure there is a full name when it firstly appears in the text.

We have checked the all abbreviations used in the manuscript and ensured that the full name appears before the abbreviation first used in the text.

  1. What equipment was used to read the Ca2+ transient signal? Also the method in this part needs more details or maybe some references are also needed, as this is helpful for readers to understand the data figure, and this is almost the most important data of the whole study.

Thank you for this reasoned comment. The equipment used to record the calcium signal and the software used to analyse the recordings are described in the text of the revised manuscript in section 2.6. In addition, we have supplemented the description of the calcium signalling analysis methodology with a method for measuring Ca2+ content in the SR and provided the source publication (Choi and Eisner, 1999; DOI: doi.org/10.1111/j.1469-7793 .1999.109ad.x), from which we developed a protocol to measure the function of the major Ca2+ transporting proteins in cardiomyocytes (SERCA2a, NCX1 and PMCA).

  1. The second sentence of section 2.6 was not grammarly correct

Thank you for drawing attention to this sentence. Parts of it in the original version of the manuscript were deleted by mistake. The sentence has been corrected.

  1. In page 8 first paragraph, it should be “RyR”

Thank you, we have provided the correct abbreviation as suggested.